# The Performance of *Miscanthus* Seeds During Long-Term Storage

**DOI:** 10.3390/plants14243738

**Published:** 2025-12-08

**Authors:** Shicheng Li, Hao Ren, Xiaoxia Huang, Zili Yi, Liang Xiao, Cheng Zheng

**Affiliations:** 1College of Bioscience and Biotechnology, Hunan Agricultural University, Changsha 410128, China; lishicheng@stu.hunau.edu.cn (S.L.); yizili889@163.com (Z.Y.); 2Hunan Crop Seed Southern Breeding Centre, Changsha 410100, China; rh36588@163.com; 3Huaihua Comprehensive Agricultural Service Centre, Huaihua 418001, China; huangxiaoxia0703@163.com; 4College of Agronomy, Hunan Agricultural University, Changsha 410128, China

**Keywords:** *Miscanthus*, seed longevity, storage condition, paraffin section

## Abstract

Seed storage is critical for preserving genetic resources, but optimal long-term storage conditions for *Miscanthus* seeds have not been established. In this five-year study, we evaluated storage protocols by comparing seed germination after four and five years, along with field establishment performance. The results demonstrated that genotype, storage conditions, and the storage duration all significantly influenced germination percentage and vigor index of *Miscanthus* seeds. Low temperature storage yielded the highest germination percentage (59.44%) and vigor index (132.06) in the 4th year, while low temperature with desiccant gave the highest germination percentage (42.41%) in the 5th year. The field performance after direct sowing was also significantly influenced by genotype and storage conditions, with the highest seedling survival (7.80%) observed under low temperature with desiccant. The seeds stored under low temperature exhibited minor structural damage, with the intact cell membranes, the small intercellular gaps, and the orderly cell arrangement. Through comprehensive evaluation, storage at −18 °C with desiccant was determined to be optimal. Based on these results, we strongly recommend storing *Miscanthus* seeds at −18 °C with desiccant. This protocol offers a reliable and effective solution for farmers, seed producers, and storage facilities to ensure long-term seed viability.

## 1. Introduction

*Miscanthus*, a C4 rhizomatous perennial grass, is recognized as a promising biomass crop [1] due to its high yield, quality, and tolerance to diverse environmental conditions [2]. As the biomass product development advances, *Miscanthus* is increasingly being used for high-value products, such as nanocellulose, rather than solely for energy generation [3]. As China focuses on carbon neutrality, the enhancement of carbon sequestration has become vital. Planting *Miscanthus* on marginal lands, such as high-salinity soils, contributes to carbon sequestration, helping mitigate climate change by reducing the atmospheric carbon levels [4,5,6]. Therefore, cultivating *Miscanthus* as an industrial crop on marginal lands not only provides a sustainable biomass source but also improves the soil quality and the carbon sequestration capacity of these lands [7,8].

The propagation of *Miscanthus* from seeds is now a reality. Compared with rhizome or in vitro propagation methods, the seed-based propagation is cost-effective, has low carbon emissions, and presents a high multiplication factor (approximately 1:2000) [9,10,11]. Through various breeding programs, new seed-based *Miscanthus* hybrids have been developed as alternatives for clonal production [12]. These hybrids, particularly interspecies hybrids, have shown the potential to match or even exceed the yields of *M. × giganteus* on lower-grade lands [13]. In the United States, field-scale seed production has been successfully achieved by synchronizing the flowering times of *Miscanthus sinensis* and *Miscanthus sacchariflorus*, significantly increasing the annual yield of viable seeds [14]. However, initial germination tests observed that *Miscanthus* seeds exhibit no dormancy and can germinate immediately after harvest. This characteristic results in spontaneous germination under favorable storage conditions, leading to a loss of seed stocks. Furthermore, germination during storage increases ambient humidity, promoting microbial growth and thereby raising the risk of seed mold and disease incidence. Patanè et al. reported that after one year of storage at room temperature, the germination percentage of *Miscanthus* seeds significantly decreases from 95.6% to approximately 60% [15]. This decline in germination percentage necessitates the use of more seeds for sowing, ultimately increasing cultivation costs.

The importance of seed storage has been recognized since plant domestication. The deterioration of seeds over time, characterized by declines in morphological structure, physiological function, and biochemical integrity [16], leads to reduced viability and germination failure [17,18]. Inappropriate storage conditions can accelerate this process, with temperature, oxygen, and relative humidity acting as critical factors [19]. For instance, the germination percentage of *Ruppia sinensis* seeds remained at 74.4% after 9 months of storage under dry conditions at 5 °C, compared to a decline to 35.7% when stored at room temperature for the same duration [20]. Currently, systematic research on storing *Miscanthus* seeds remains limited. As *Miscanthus* gains prominence, developing an economical, efficient, and scalable seed storage system has become an urgent requirement for supporting its industrial expansion.

This study designed six distinct storage methods for *Miscanthus* and evaluated their effectiveness by evaluating the germination percentages in the 4th and 5th years of storage, as well as by conducting field experiments in the 5th year. The changes in the cellular level were also observed. This study assessed the viability of *Miscanthus* seeds after long-term storage under various conditions, with the objective of developing an effective storage method to support future large-scale *Miscanthus* seed cultivation.

## 2. Result

### 2.1. Germination Performance of Miscanthus Seed During Storage

Figure 1 illustrates the cumulative germination percentages of the six seed genotypes after the long-term storage under four different conditions. The seeds stored at room temperature and those stored at room temperature under vacuum conditions failed to germinate in the fourth and fifth years, and their cumulative germination percentages are not depicted in Figure 1. The differences in the initial germination dynamics were observed between stored and unstored seeds. For example, the unstored *M. lutarioriparius* seeds (Y0101 and B0129) demonstrated the fastest germination percentages, with peak germination occurring on the 3rd and 4th days, respectively. In contrast, the other four genotypes required 6–8 d to reach their maximum germination percentage.

### 2.2. Effect of Storage Conditions on Seed Germination and Vigor Index Under Long-Term Storage

Initial germination tests revealed considerable variation among genotypes, with germination percentages ranging from 70.67% to 92.00% and vigor index from 125.25 to 331.27 (Appendix A). Both storage duration and genotype had significant effects on *Miscanthus* seed germination percentage and vigor index (Table 1). Germination percentage and vigor index declined significantly over time (*p* < 0.05; Figure 2). After 4 years of storage, the mean germination percentage and vigor index dropped to 30.99% and 60.77, respectively, further declining to 23.09% and 37.87 after five years. They were both significantly lower than in newly harvested seeds. B0129 maintained the highest average germination percentage in both the 4th (47.22%) and 5th (42.04%) years (Figure 3), whereas D0115 showed the lowest in the 4th year (19.26%) and C0615 the lowest in the 5th year (6.67%). Storage conditions significantly influenced germination percentage and vitality index of *Miscanthus* seeds (Table 1). Seeds under all three low-temperature conditions consistently outperformed those stored at room temperature. A significant interaction between storage conditions and storage duration was observed (Table 1). Low temperature storage yielded the highest germination percentage (59.44%) and vigor index (132.06) in the 4th year, while low temperature with desiccant gave the highest germination percentage (42.41%) in the 5th year. Seeds stored at room temperature with desiccant exhibited the most severe decline, with losses of 81.61% in germination percentage and 94.07% in vigor index after five years. In contrast, seeds stored at low temperature with desiccant showed the smallest loss in germination percentage (48.63%), while those under low temperature exhibited the smallest reduction in vigor index (60.66%). Furthermore, during the late storage stage (years 4 to 5), the low temperature with vacuum condition was the most effective, minimizing the annual reduction in both germination percentage (12.60%) and vigor index (28.62%).

### 2.3. Field Seed Germination and Survival

All seeds stored under different conditions exhibited low emergence percentages and survival percentages when planted in the field during the 5th year of storage. Appendix A presents the emergence percentages, survival, shoot lengths, and dry biomass measurements of *Miscanthus* seeds from the field experiments. The emergency percentages in the field ranged from 0% to 17.78% across the six genotypes stored under the six conditions, with significant differences observed (Table 2). No emergence was observed in seeds stored at room temperature or under vacuum at room temperature. Although some seeds stored at room temperature with desiccant initially emerged, none survived. All seeds stored at low temperatures with desiccant conditions successfully emerged in the field, with an average emergence percentage of 8.15%. The seeds stored at low temperature and those stored at low temperature with vacuum conditions demonstrated emergence in all genotypes except C0615, with average emergence percentages of 6.85% and 5.74%, respectively. Among all storage conditions, only three genotypes (B0129, Y0101, and Z0101) survived in the field under all three low-temperature storage treatments. Among these, no significant differences were observed in either emergence or survival percentages across the three low-temperature conditions (Figure 4B). Furthermore, a positive correlation was found between the laboratory germination percentage and the field survival after 5 years of storage (Appendix A). A general linear model analysis of the shoot length and dry weight of the surviving seedlings revealed that only the genotype significantly affected shoot length (*p* < 0.05). The genotype Y0101 exhibited the highest shoot length (49.40 cm) and dry biomass weight (1.49 g).

### 2.4. Ranking of Species and Storage Conditions by Membership Function Analysis

Figure 5 displays comprehensive results based on the membership values, which are composite scores derived from a weighted evaluation of multiple seed vigor traits. The composite evaluation scores among species were as follows: Hybrid (0.99), *M. lutarioriparius* (*M. lut*) (0.80), *M. sinensis* (*M. sin*) (0.14), and *M. sacchariflorus* (*M. sac*) (0). The survival of the hybrid in the field was higher than that of the other species after five years of storage. The germination percentage of the Hybrid and *M. lut* were similar and higher than that of the other species. *M. sac* exhibited poor storage tolerance, with the lowest germination percentage and the survival after five years. The composite evaluation scores among the storage conditions were as follows: low temperature with desiccant (0.99), low temperature with vacuum (0.88), low temperature (0.82), room temperature with desiccant (0.12), room temperature with vacuum (0), and room temperature (0). The survival of *Miscanthus* seeds was highest under the low-temperature desiccant conditions. Among the three low-temperature conditions, the germination percentages of *Miscanthus* seeds were similar and significantly higher than those under the three room temperature conditions.

### 2.5. Paraffin Section of Long-Term Stored Miscanthus Seeds

To assess the impact of storage conditions on the internal structure of *Miscanthus* seeds, the seeds stored for five years under room temperature and low temperature conditions were subjected to paraffin sectioning experiments (Figure 6). The seeds stored at low temperatures exhibited minimal damage to their microstructure. The embryo and endosperm were full, with small gaps between them and the seed coat. The cell membranes of embryo and endosperm cells remained intact, with small intercellular gaps and neatly arranged cells (Figure 6A,B). In contrast, the seeds stored at room temperature exhibited significant structural damage (Figure 6C,D). The distance between the embryo, endosperm, and seed coat increased, and the incomplete cell membranes and partial tissue loss were observed in both the embryo and endosperm. The gap between the embryo and endosperm also increased, and the arrangement of cells at their junction was relatively disordered. These observations indicated that the seed cell membranes were damaged and that the structural stability was compromised during storage at room temperature. Additionally, the connection between the embryo and endosperm weakened.

## 3. Discussion

### 3.1. Miscanthus Seed Dormancy Failure and Short Longevity

*Miscanthus* seeds do not exhibit dormancy. In 2016, immediately after the *Miscanthus* seeds reached maturity, we removed the glumes and promptly initiated germination experiments. The results indicated that the seeds began to germinate rapidly on the second day of the experiment, with a notably high germination percentage. The seed dormancy was defined as the condition under which viable seeds failed to germinate under the optimal temperature, humidity, and oxygen conditions [21]. This lack of dormancy was disadvantageous for preserving *Miscanthus* seeds, as they germinated immediately upon encountering suitable environmental conditions, leading to a reduction in seed numbers. Therefore, it could be essential to dry and store *Miscanthus* seeds immediately after harvest to prevent losses due to spontaneous germination.

*Miscanthus* seeds typically have a short lifespan, attributable not only to their non-dormancy characteristics but also to their inherent genetic traits. The average weight of a thousand *Miscanthus* seeds was approximately 0.45 g, and their small size limits the available nutrients for consumption. As nutrients were depleted during storage, supporting successful germination has become increasingly challenging. Generally, the *M. lutarioriparius* seeds and hybrid seeds with *M. lutarioriparius* as the female parent were the largest in volume, consistent with the observations of this experiment. Specifically, the seeds from *M. lutarioriparius* (B0129 and Y0101) and the hybrid (Z0101) demonstrated superior germination performance after long-term storage compared to other species, and they also demonstrated stronger vitality in field trials.

### 3.2. Impact of Storage Conditions on Miscanthus Seed Longevity

In line with the results of this study and those of Meyer et al. [22], *Miscanthus* seeds can be classified as orthodox, indicating that reducing the relative humidity and temperature of the storage environment benefits their longevity [23]. The use of desiccants effectively preserved the vitality of *Miscanthus* seeds at room temperature. In contrast, the other two room temperature storage methods, such as room temperature alone and room temperature with vacuum, failed to support germination in the 4th and 5th years. The successful germination of seeds stored at room temperature with desiccant is likely attributable to the maintenance of low relative humidity, which retards seed aging and degradation. The final moisture content of seeds during storage is influenced by relative humidity, as seeds reach equilibrium with the surrounding water vapor [24]. Higher relative humidity increases seed moisture content, which accelerates seed aging and reduces germination percentage [25]. The experimental site, Changsha, has a subtropical monsoon climate with a relative humidity exceeding 50% year-round. The seeds stored at room temperature are continually exposed to high relative humidity, leading to accelerated aging. Using desiccants, a consistently low and stable relative humidity environment was maintained, allowing the *Miscanthus* seeds to remain viable even after 5 years of storage at room temperature.

Reducing the temperature was more effective for the long-term preservation of *Miscanthus* seeds. Under the three different low-temperature storage conditions, *Miscanthus* seeds maintained higher vitality. Typically, the effect of low temperature desiccants was similar to that of low temperature alone, as desiccants were ineffective at −18 °C. However, during long-term storage, unforeseen power outages may occur, causing the temperature and relative humidity to rise inside the refrigerator. In such cases, the desiccant helps to mitigate the adverse effects of these environmental changes by providing emergency protection. Although the vitality index of the seeds stored under the low temperature conditions was slightly higher than that of seeds stored under the low temperature with desiccant conditions, the difference was not significant and may be due to the inherent seed variability. Overall, the low-temperature storage consistently proved more effective in maintaining the seed vitality than the room-temperature storage.

For *Miscanthus* seeds, vacuum storage demonstrated a limited effect on maintaining seed viability. The vacuum treatment, isolating the moisture and oxygen, failed to preserve *Miscanthus* seeds during long-term storage, as seeds at room temperature with vacuum lost viability by the fourth year. Moreover, the germination percentage and vitality index of the seeds stored under the low temperature with vacuum conditions were the lowest among the three low-temperature storage methods. This result contrasts sharply with the benefits of vacuum storage reported in other species. For instance, Flores et al. reported that the vacuum treatment prevented a 50% reduction in the germination capacity of *Juglans nigra* L. seeds stored at −20 °C for one year [26]. Similarly, Meena et al. observed that vacuum packaging extended the shelf life of soybean seeds to 18 months without compromising their quality indicators [27]. Furthermore, Barzali et al. discovered a positive effect on the germination percentage of *Secale graale* L. seeds under the room temperature vacuum conditions during a 26-year storage experiment [28]. Hence, the effectiveness of vacuum storage is highly dependent on species. For seeds such as those of *Miscanthus*, the vacuum environment may not adequately suppress aging processes, or the seeds themselves might be more sensitive to the vacuum conditions. Therefore, when formulating storage protocols, it is crucial to fully consider species-specific seed characteristics, rather than treating vacuum storage as a universally applicable optimization method.

### 3.3. Field Performance of Miscanthus Seeds After Long-Term Storage

To evaluate the field germination of the long-stored *Miscanthus* seeds, we sowed the seeds directly without prior treatment. The results were unsatisfactory, consistent with previous studies that highlighted that the direct sowing of *Miscanthus* seeds was difficult to establish successfully [29,30]. For field planting, a “two-step” seed propagation method is recommended for *Miscanthus* over direct sowing [31]. This approach involves first cultivating seedlings in nursery beds, then strengthening them in plug trays to produce robust plantlets suited to field conditions. The seeds stored at room temperature with desiccant did not survive despite the emergence of some seedlings. The seeds stored under the three low-temperature conditions generally emerged, but only those of *M. lutarioriparius* (B0129 and Y0101) and the hybrid *M. lutarioriparius* × *M. sinensis* (Z0101) survived. *M. lutarioriparius* and its hybrid varieties proved to be excellent biomass varieties within the *Miscanthus* genus, exhibiting higher biomass yields [32]. The higher seed viability of *M. lutarioriparius* and its hybrids compared to other *Miscanthus* species could provide a significant advantage in selecting *M. lutarioriparius* for promotion.

### 3.4. Effect of Long-Term Storage on the Cell Structure of Miscanthus Seeds

The embryo derived from a fertilized egg is the most crucial component of the seed and represents the young plant body. The cellular structure of the embryo serves as a key indicator of the physiological activity of the seed and undergoes changes as the seed ages. Among the environmental factors, relative humidity and temperature have the most direct influence on seed aging. This experiment utilized two distinct relative humidity and temperature conditions (room temperature and low temperature) for paraffin sectioning of *Miscanthus* seeds. The seed cell structures demonstrated the notable differences under these conditions. The seeds stored at low temperatures maintained an intact cell structure with no apparent signs of aging, whereas the seeds stored at room temperature exhibited significant structural damage, including incomplete cell membranes and partial tissue loss in both the embryo and endosperm. As the seeds aged, the plasma membrane deteriorated, primarily due to lipid peroxidation [33]. Increased aging led to higher concentrations of reactive oxygen species (ROS), which attacked the polyunsaturated fatty acids in membrane phospholipids. This process resulted in the breakdown of the long-chain fatty acids into smaller compounds, altering membrane permeability and causing membrane damage [18]. Consequently, the primary cause of the failure of *Miscanthus* seeds stored at room temperature to germinate was seed aging and destruction of the cellular structure.

### 3.5. Optimal Conditions for Storing Miscanthus Seeds

To evaluate the impact of various storage conditions on *Miscanthus* seeds, we employed a membership function. This study aimed to provide practical guidance for the application of *Miscanthus* seeds, with the particular emphasis on the field survival, which has been assigned the highest weight. Although the field survival results were not entirely satisfactory, they revealed significant differences among the different storage conditions and genotypes. Additionally, the germination percentage, a primary indicator of seed quality, was also given substantial weight in the evaluation.

The results of the comprehensive evaluation indicated that the low temperature with desiccant storage was optimal for maintaining *Miscanthus* seeds’ vitality. This finding is significant for both seed production and long-term preservation, demonstrating that high seed vitality can be sustained using a relatively simple storage method. Although storage at −18 °C with desiccant incurs higher energy costs than other storage, it is strongly recommended for protecting high-value *Miscanthus* genetic resources, breeding lines, and commercial seed stocks. Producing new *Miscanthus* seeds is a prohibitively expensive and lengthy process, requiring a full year of dedicated land use, controlled pollination, labor-intensive harvesting, and post-harvest processing. By contrast, the energy cost of maintaining a −18 °C freezer is relatively modest. Moreover, the desiccant is not a consumable but a reusable resource. It can be regenerated by drying, thereby distributing its cost over many years. Consequently, while storage at −18 °C with desiccant has an ongoing cost, it is overwhelmingly cost-effective compared to the alternative of frequent seed regeneration. This makes the −18 °C with the desiccant method not only biologically superior but also economically imperative. To ensure the sustained viability of *Miscanthus* seeds, a routine monitoring program is critical. For seeds stored at −18 °C with desiccant, we recommend annual germination testing to verify the maintenance of high viability.

The evaluation also revealed the differences among genotypes. Seeds from *M. lutarioriparius* and its hybrids, with *M. lutarioriparius* as the maternal parent, exhibited superior performance, whereas *M. sacchariflorus* seeds performed the worst. Therefore, *M. lutariparius* should be prioritized as the female parent in breeding new *Miscanthus* varieties to cultivate strains with higher vitality and adaptability.

## 4. Conclusions

To promote the cultivation of *Miscanthus* in marginal lands, there is a significant need for *Miscanthus* resources. Seed propagation could offer a rapid and cost-effective alternative to the high costs of traditional rhizome propagation. However, because *Miscanthus* seeds lack a dormancy period, identifying the suitable storage conditions is crucial. This study developed an economical and straightforward storage method through long-term experiments: drying the *Miscanthus* seeds and storing them with desiccants at −18 °C to maintain the long-term viability. The seeds stored under these conditions for five years achieved the maximum average germination percentage of 75.56%, although the maximum average field survival was relatively low at 8.89%. Thus, improving the field survival remained necessary. Additionally, this study observed a significant impact of genotypes on *Miscanthus* seed storage. However, owing to the limited number of species studied (two *Miscanthus sinensis*, two *Miscanthus lutarioriparius*, one *Miscanthus sacchariflorus*, and one interspecific hybrid of *Miscanthus sinensis* and *Miscanthus lutarioriparius*), the effects of different *Miscanthus* species on seed storage are not fully understood.

## 5. Materials and Methods

### 5.1. Origin and Storage Conditions of Miscanthus Seeds

The *Miscanthus* seeds utilized in this study were obtained from the *Miscanthus* Resource Garden at Hunan Agricultural University in Hunan Province, China (28°11′ N, 113°4′ E). Seeds were collected from six maternal lines (Appendix A) grown in this common garden, where natural hybridization could occur. All subsequent references to species and hybrids are based on these known maternal lines. The lines included two *Miscanthus sinensis*, two *Miscanthus lutarioriparius*, one *Miscanthus sacchariflorus*, and one interspecific hybrid (*M. sinensis* × *M. lutarioriparius*). The seeds were harvested on 25 December 2016, and were then subjected to treatment and preliminary storage following the protocol developed by Chau et al. [34]. Subsequently, the seeds were randomly selected for the pre-treatment, which included (1) assessing the initial seed viability after collection, (2) drying the seeds with desiccants, and (3) testing the seed vitality after drying. Freshly harvested seeds had a moisture content of 18.62%, compared to 12.15% after drying. This was measured by the oven method at 105 °C until constant weight was achieved. The initial tests indicated that the *Miscanthus* seeds could germinate in the second test, suggesting that the *Miscanthus* seeds were not recalcitrant. Therefore, storing the seeds under oxygen-rich, warm, and moist conditions, which were typically required for recalcitrant seeds, could be unnecessary.

The dry seeds were placed into the centrifuge tubes with small holes in the top cover and subjected to six different storage conditions: (1) room temperature (RT), (2) room temperature with desiccant (RTD), (3) room temperature with vacuum (RTV), (4) low temperature (LT), (5) low temperature with desiccant (LTD), and (6) low temperature with vacuum (LTV) (Appendix A). To achieve the desired drying conditions, the centrifuge tubes containing the seeds were placed in Falcon tubes filled with silica gel pellets. For the vacuum treatment, the tubes were packed and sealed in aluminum foil bags using a vacuum sealer. Then they are transferred to a freezer at −18 °C or stored at room temperature. All the seeds were kept in the dark.

Temperature (°C) and relative humidity (%) data for the experimental site were obtained from https://wheata.cn/ (accessed on 17 January 2024). Detailed monthly fluctuations under the ambient condition are provided in Appendix A. Room temperature storage conditions entailed exposure to the ambient environment of the experimental site. The mean temperature ranged from 4.6 °C to 30.8 °C. Maximum temperatures commonly occurred in July and August, with peaks reaching 40.0 °C, while minimum temperatures were recorded in January and February, dropping to as low as −5.0 °C. Relative humidity was consistently high, exceeding 60% in all months except September 2019, when it dropped to 51.56%. The maximum relative humidity recorded was 89.61%.

### 5.2. Performance of Seed Germination During Long-Term Storage

The seed germination experiments were conducted under consistent conditions to assess the viability of the seeds after long-term storage. To prevent fungal infestation during the germination test, the seeds were disinfected with the sodium hypochlorite solution and then rinsed with distilled water. The germination experiments used the Petri dishes with the filter paper as the growth medium, and the Petri plates were sterilized by autoclaving. The samples were placed evenly on the double-layer filter paper in the culture dishes, which were then incubated in the seed incubator at 25 ± 0.5 °C with alternating light (12 h light/dark photoperiod). They were maintained in a moist environment with the daily application of distilled water. Each treatment per genotype included 30 seeds and was replicated three times to ensure statistical reliability [35]. Germination was assessed by the presence of a visible radicle, and the germinated seeds were counted daily. The experiment was terminated if no new seeds were germinated for three consecutive days. Germination experiments were conducted in 2016, 2020 and 2021, with a focus on the later stages of storage when a substantial loss of viability was expected, to assess the long-term effectiveness of the storage methods.

### 5.3. Performance of Seed Establishment in the Field After Long-Term Storage

Field trials were conducted in late July 2021 at Hunan Agricultural University to evaluate the practical field establishment potential of seeds after five years of storage. The primary objective was to determine whether the viability observed in laboratory tests after extended storage translated into successful field establishment. This assessment served as the definitive test of the storage method’s practical effectiveness. The designated area was prepared for seed germination, which was divided into 108 plots, encompassing six genotypes, six storage methods, and three replications. Prior to the trial, the vegetation and litter were removed from the plots. The stored seeds from each group were randomly distributed on the plot surface, with each group consisting of 30 seeds. The regular watering every 2 d ensured the seed germination and growth. The germinated seeds were counted on the 10th day of sowing. After 60 d, all the surviving seedlings were collected, and the survival, shoot length, and dry biomass weight were measured. To determine the dry biomass weight, the seedlings were placed in a desiccator at 65 °C until a constant mass was achieved and then weighed. During the field experiment, the average highest temperature was 34.25 °C, the average lowest temperature was 25.72 °C, and total precipitation was 144.10 mm.

### 5.4. Paraffin Section of Long-Term Storage of Miscanthus Seeds

The paraffin sectioning was performed to examine the morphological changes in *Miscanthus* seeds after storage. To clearly reveal the differences in the internal structure of the seeds, we selected the seeds stored under the conditions with the highest and lowest germination percentages for these experiments. The seeds were initially rinsed with distilled water and subsequently immersed in a fixative composed of formaldehyde, acetic acid, and 50% ethanol for 24 h. Following the fixation, the seeds were subjected to a dehydration process using a graded series of ethanol solutions: 50% ethanol for 1 h, 70% for 2 h, 85% for 2 h, 95% for 1 h, and two changes of 100% ethanol for 45 min each. To achieve tissue transparency, 100% ethanol was replaced sequentially by a 1:1 mixture of 100% ethanol–dimethylbenzene, followed by dimethylbenzene, with each step lasting 1 h. The dimethylbenzene was then replaced with the 1:1 mixture of dimethylbenzene and wax, followed by three treatments with pure wax, each involving incubation in an oven at 56 °C for 4 h. The molten wax was poured into the embedding boxes, and the material was adjusted using a dissecting needle to facilitate sectioning. For the structural observations, the sections were cut using a rotary microtome and floated on the surface of a water bath set at 40 °C to flatten them. The sections were carefully transferred to the microscope slides and vertically placed in a slide holder. The slide holder was then dried in an oven at 42 °C to ensure the proper adhesion of the sections. Subsequently, the paraffin sections were deparaffinized through two treatments with xylene, each lasting 20 min. The slides were immersed in a graded ethanol series (xylene–ethanol 1:1; 100%, 95%, 85%, and 70% ethanol for 30 min each). The staining was performed using Safranine O-Fast Green Stain, according to the manufacturer’s standard protocol (Solarbio Life Science, Beijing, China). Multiple seed samples were prepared and imaged microscopically. The micrographs shown are representative of the observations.

### 5.5. Statistical Analysis

All experimental data were analyzed using the general linear model (GLM) in Origin (version 2025) software. The statistical significance of differences between means was assessed using the Duncan test, with significant results reported at the *p* < 0.05 level. Given that a single trait could often fail to fully reflect the storage performance of *Miscanthus* seeds, this study employed a membership function approach for a comprehensive evaluation of seeds stored for five years. This method normalizes the original data of each seed vigor trait to a relative scale from 0 (poorest performance) to 1 (best performance), resulting in a dimensionless membership value for each trait. To address the varying importance of different traits, a weighted sum of the membership values was then calculated. The field survival, as a decisive indicator of crop establishment success, was assigned a weight of 0.6. The indoor germination percentage, the primary measure of seed quality, was assigned a weight of 0.3. The remaining four traits were collectively assigned a weight of 0.1 to reflect their supportive role in the overall evaluation system. The final composite evaluation score was derived from these weighted membership values.

## Figures and Tables

**Figure 1 plants-14-03738-f001:**
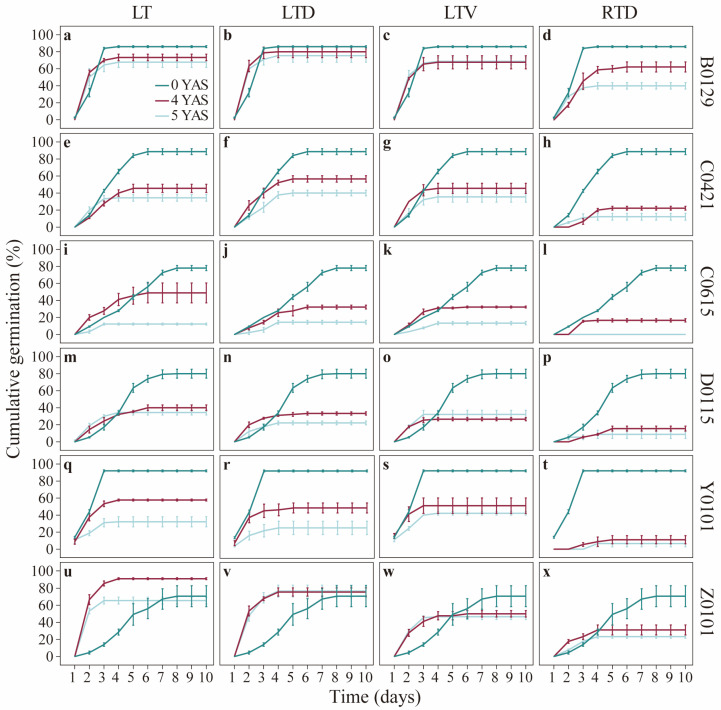
Cumulative germination of seeds. Figures show seeds of (**a**–**d**) B0129, (**e**–**h**) C0421, (**i**–**l**) C0615, (**m**–**p**) D0115, (**q**–**t**) Y0101, and (**u**–**x**) Z0101, stored under conditions LT, LTD, LTV and RTD, respectively. B0129, C0421, C0615, D0115, Y0101 and Z0101 refer to the codes of the distinct maternal plants used in the study. None of the seeds from all tested lines germinated after 4 and 5 years of storage under room temperature and room temperature with vacuum conditions; therefore, they were omitted. RTD, seeds were stored under room temperature with desiccant condition; LT, seeds were stored under low temperature condition with no treatment; LTD, seeds were stored under low temperature with desiccant condition; LTV, seeds were stored under low temperature with vacuum condition.

**Figure 2 plants-14-03738-f002:**
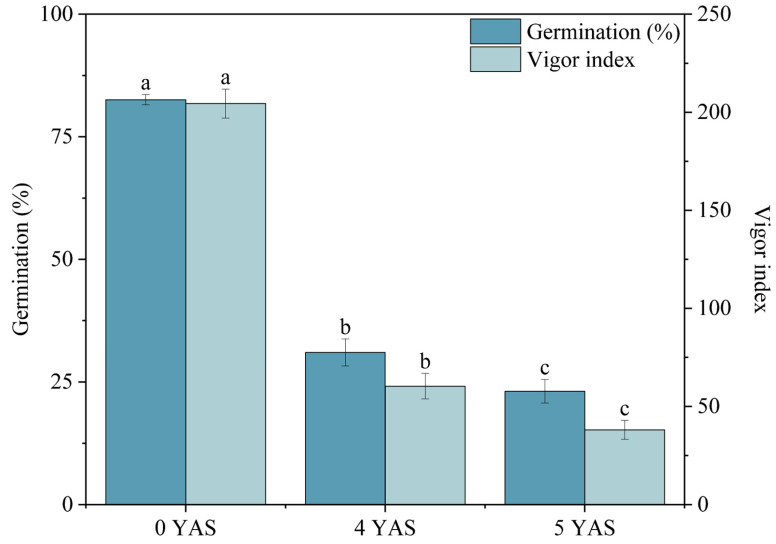
Posterior comparisons of germination percentage and vigor index of storage duration. Error bars on the mean indicate standard errors. Different lowercase letters represent statistical significance (*p* < 0.05). YAS, years after storage.

**Figure 3 plants-14-03738-f003:**
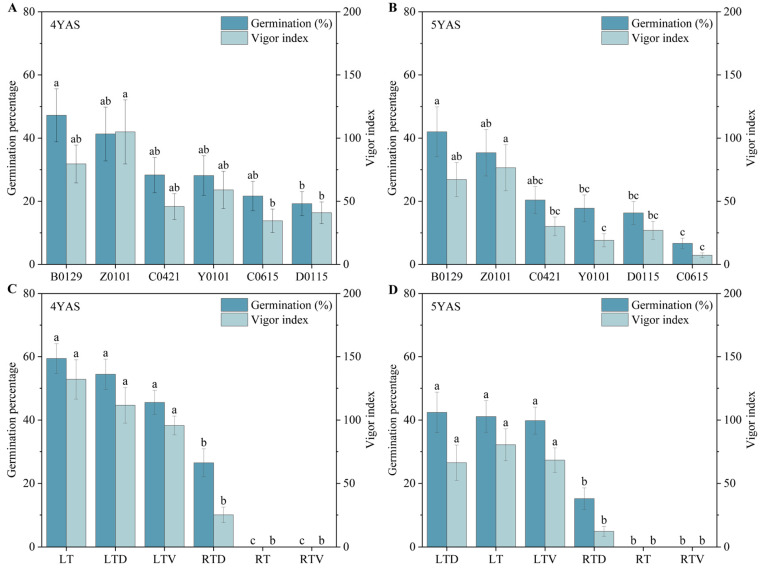
Posterior comparisons of germination percentage and vigor index of genotypes (**A**,**B**) and storage conditions (**C**,**D**). Error bars on the mean indicate standard errors. Different lowercase letters represent statistical significance (*p* < 0.05). YAS, years after storage. RT, seeds were stored under room temperature condition without treatment; RTD, seeds were stored under room temperature with desiccant condition; RTV, seeds were stored under room temperature with vacuum condition; LT, seeds were stored under low temperature condition with no treatment; LTD, seeds were stored under low temperature with desiccant condition; LTV, seeds were stored under low temperature with vacuum condition.

**Figure 4 plants-14-03738-f004:**
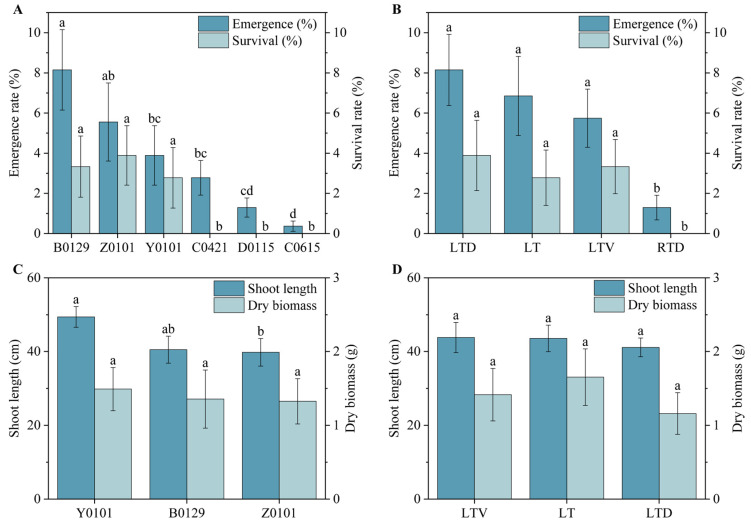
Posterior comparisons of seed characteristics in field experiments. (**A**) Emergence and survival percentages among genotypes. (**B**) Emergence and survival percentages among storage conditions. (**C**) Shoot length and dry biomass of genotypes. (**D**) Shoot length and dry biomass of storage conditions. Error bars on the mean indicate standard errors. Different lowercase letters represent statistical significance (*p* < 0.05). RT, seeds were stored under room temperature condition without treatment; RTD, seeds were stored under room temperature condition with desiccant; RTV, seeds were stored under vacuum, room temperature condition; LT, seeds were stored under low temperature condition without treatment; LTD, seeds were stored under low temperature condition with desiccant; LTV, seeds were stored under vacuum, low temperature condition.

**Figure 5 plants-14-03738-f005:**
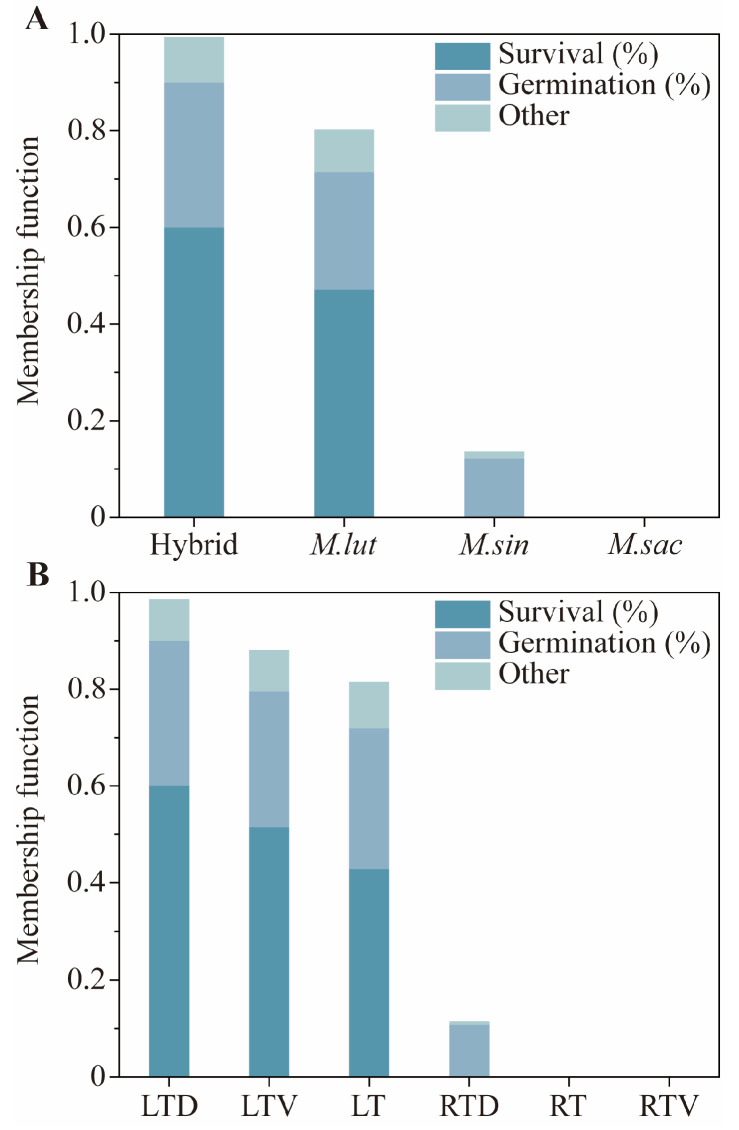
Membership values of species (**A**) and six storage conditions (**B**). Other, the composite membership value for vigor index, emergence percentage, shoot length, and dry biomass. RT, seeds were stored under room temperature condition without treatment; RTD, seeds were stored under room temperature condition with desiccant; RTV, seeds were stored under room temperature with vacuum condition; LT, seeds were stored under low temperature condition without treatment; LTD, seeds were stored under low temperature condition with desiccant; LTV, seeds were stored under low temperature with vacuum condition.

**Figure 6 plants-14-03738-f006:**
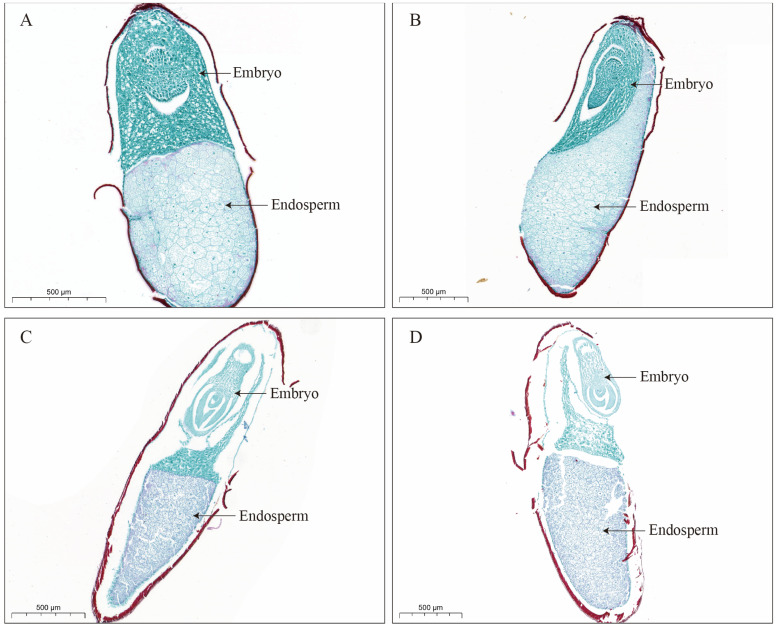
Microscopic structure of *Miscanthus* seeds after 5 years of storage. (**A**,**B**) Storage at low temperature; (**C**,**D**) storage at room temperature.

**Table 1 plants-14-03738-t001:** General linear model (GLM) of traits related to germination percentage and vigor index of seeds.

Storage Factors	Germination Percentage (%)		Vigor Index	
	*F*-Value	*p*-Value	*F*-Value	*p*-Value
Genotype (G)	52.869	0.000	270.693	0.000
Storage condition (SC)	364.995	0.000	269.546	0.000
Storage duration (SD)	1993.599	0.000	4230.262	0.000
G × SC	6.725	0.000	16.871	0.000
G × SD	22.428	0.000	220.029	0.000
SC × SD	95.362	0.000	78.165	0.000
G × SC × SD	2.264	0.000	5.868	0.000

**Table 2 plants-14-03738-t002:** General linear model (GLM) for seed emergence percentage, survival, shoot length, and dry biomass.

Trait	Genotype (G)		Storage Condition (SC)		G × SC	
	*F*-Value	*p*	*F*-Value	*p*	*F*-Value	*p*
Emergence percentage	10.00	0.00	24.79	0.03	1.80	0.03
Survival	5.12	0.00	4.98	0.00	1.31	0.19
Shoot length	3.57	0.04	0.85	0.43	2.01	0.06
Dry biomass	0.32	0.73	0.73	0.49	1.64	0.18

## Data Availability

The data that support the findings of this study are available from the corresponding authors upon reasonable request.

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
