# Peer review of "The Performance of *Miscanthus* Seeds During Long-Term Storage"

_plants, 2025, doi:10.3390/plants14243738_

Round 1
Reviewer 1 Report
Comments and Suggestions for Authors
This paper presents results comparing different seed storage conditions on seed germination and seedling establishment of Miscanthus, a popular biomass crop, identifying a recommended condition. This paper presents a minor contribution to seed viability research that would interest researchers, suppliers, and growers of miscanthus and other similar species. The introduction provides suitable referenced background to the study system and identifies a knowledge gap around recommended seed storage conditions for this species. The aims and approaches are clearly presented at the end of the introduction. The results are explained clearly and summarised well with relevant tables and figures. It would be better to have some indication of sample size here for these analyses. Including degrees of freedom estimates in the GLM tables could help here. Comparisons between Miscanthus species/hybrids needs to be mentioned earlier in the study aims as one of the study aims. Later membership results do not make sense in advance of the methods section and need to be briefly explained in the results section. The microscopy images should include labels of different features. The discussion is structured well to develop different aspects of the results and linking these interpretations with the related literature. The methods are well structured and concisely describe the approaches with sufficient detail for repeatability. The seed sources described in the methods introduce some doubt about the identity of each tested line that needs to be clarified. The conclusions end the paper well with some overview perspectives and ideas for future study. Overall the study is presented well and the results will be of some interest to Miscanthus breeders and growers and others in the seed storage industry more generally.
Specific comments
L93-95 I do not see this general observation of faster germination with prolonged storage. Lines C0241 and Y0101 seem to show the opposite pattern.
L97-103 Explain the row names B0129, etc. and the RT, RTV, treatment acronyms.
L149 Figure 3 not cited in the main text.
L198 "Comprehensive evaluation" is a vague heading title. Better to state what specific results are summarised in this section.
L200-201 Spell out species acronyms at first use.
L205-208 The meaning of membership values is unclear and needs to be explained here before the methods.
L214 "Other" category needs to be explained.
L237 The different seed and embryo structures of interest should be indicated with arrows.
L313-315 Explain the two-step approach here.
L363-365 What is the confidence that the seed comes only from that genotype and not hybridization if the lines were growing together in a garden? If their lineage cannot be guaranteed, it would be better to only state that the maternal line was known.
L372-373 Briefly state how seed moisture content was measured.
L446 State how many seeds per line were examined.
English language needs some checking and editing for overuse of "the".
Author Response
Reviewer #1:
This paper presents results comparing different seed storage conditions on seed germination and seedling establishment of Miscanthus, a popular biomass crop, identifying a recommended condition. This paper presents a minor contribution to seed viability research that would interest researchers, suppliers, and growers of miscanthus and other similar species. The introduction provides suitable referenced background to the study system and identifies a knowledge gap around recommended seed storage conditions for this species. The aims and approaches are clearly presented at the end of the introduction. The results are explained clearly and summarised well with relevant tables and figures. It would be better to have some indication of sample size here for these analyses. Including degrees of freedom estimates in the GLM tables could help here. Comparisons between Miscanthus species/hybrids needs to be mentioned earlier in the study aims as one of the study aims. Later membership results do not make sense in advance of the methods section and need to be briefly explained in the results section. The microscopy images should include labels of different features. The discussion is structured well to develop different aspects of the results and linking these interpretations with the related literature. The methods are well structured and concisely describe the approaches with sufficient detail for repeatability. The seed sources described in the methods introduce some doubt about the identity of each tested line that needs to be clarified. The conclusions end the paper well with some overview perspectives and ideas for future study. Overall the study is presented well and the results will be of some interest to Miscanthus breeders and growers and others in the seed storage industry more generally.
Ans: Thank you very much for giving us this valuable opportunity to revise. Your review comments are very pertinent and instructive, providing us with valuable directions. We have carefully revised the manuscript according to your suggestions. The revised version has been submitted, with all changes highlighted in red for your convenience. Additionally, we have made further improvements beyond the specific comments to enhance the manuscript's quality. We kindly ask for your review of the revised version.
Specific comments
L93-95 I do not see this general observation of faster germination with prolonged storage. Lines C0241 and Y0101 seem to show the opposite pattern.
Ans: Thank you for your valuable suggestions. We agree with the reviewer's observation. The statement regarding faster germination with prolonged storage has been removed from the manuscript, as it was not universally supported by the data.
L97-103 Explain the row names B0129, etc. and the RT, RTV, treatment acronyms.
Ans: Thank you very much for your reminder. The explanations for the row names (B0129, etc., representing maternal line codes) and treatment acronyms have been added.
Page 5, lines 92-95
“Figure 1. Cumulative germination of seeds. B0129, C0421, C0615, D0115, Y0101 and Z0101 refer to the codes of the distinct maternal plants used in the study. None of the seeds from all tested lines germinated after 4 and 5 years of storage under room temperature and room temperature with vacuum conditions; therefore, they were omitted.”
L149 Figure 3 not cited in the main text.
Ans: Thank you very much for your reminder. Figure 3 has been cited in the revised manuscript as suggested.
Page 6, lines 108-111
“B0129 maintained the highest average germination percentage in both the 4th (47.22%) and 5th (42.04%) years (Figure 3), whereas D0115 showed the lowest in the 4th year (19.26%) and C0615 the lowest in the 5th year (6.67%).”
L198 "Comprehensive evaluation" is a vague heading title. Better to state what specific results are summarised in this section.
Ans: Thank you very much for your suggestion. The heading has been changed to more specifically reflect the section's content.
Page 11, lines 177
“2.4 Ranking of species and storage conditions by membership function analysis”
L200-201 Spell out species acronyms at first use.
Ans: Thank you for your kind reminder. The species acronyms have now been spelled out in full at their first use in the revised manuscript.
Page 12, lines 180-182
“The composite evaluation scores among species were as follows: Hybrid (0.99), M. lutarioriparius (M. lut) (0.80), M. sinensis (M. sin) (0.14), and M. sacchariflorus (M. sac) (0).”
L205-208 The meaning of membership values is unclear and needs to be explained here before the methods.
Ans: We thank the reviewer for this suggestion. As recommended, we have now provided an explanation of the membership values in the Results section. The explanation clarifies that these values are composite scores derived from a weighted evaluation of multiple traits.
Page 11, lines 178-179
“Figure 5 displays comprehensive results based on the membership values, which are composite scores derived from a weighted evaluation of multiple seed vigor traits.”
L214 "Other" category needs to be explained.
Ans: Thank you for your valuable suggestions. The 'Other' category in Figure 5 has been defined in the figure caption as the composite membership value for vigor index, emergence percentage, shoot length, and dry biomass.
Page 13, lines 194-195
“Other, the composite membership value for vigor index, emergence percentage, shoot length, and dry biomass.”
L237 The different seed and embryo structures of interest should be indicated with arrows.
Ans: Thanks for your reminder. As suggested, we have updated Figure 6 by adding arrows to indicate the embryo and endosperm.
L313-315 Explain the two-step approach here.
Ans: Thank you for your suggestion. As recommended, we have now added an explanation of the "two-step" approach in the revised manuscript.
Page 18, lines 297-300
“For field planting, a “two-step” seed propagation method is recommended for Miscanthus over direct sowing [31]. This approach involves first cultivating seedlings in nursery beds, then strengthening them in plug trays to produce robust plantlets suited to field conditions.”
L363-365 What is the confidence that the seed comes only from that genotype and not hybridization if the lines were growing together in a garden? If their lineage cannot be guaranteed, it would be better to only state that the maternal line was known.
Ans: Thank you for your valuable suggestions. The text has been revised to clarify that the study is based on seeds collected from distinct maternal lines, and all references to species or hybrids are based on this known maternal lineage.
Page 21, lines 375-379
“Seeds were collected from six maternal lines (Table S3) grown in this common garden, where natural hybridization could occur. All subsequent references to species and hybrids are based on these known maternal lines. The lines included two Miscanthus sinensis, two Miscanthus lutarioriparius, one Miscanthus sacchariflorus, and one interspecific hybrid (M. sinensis × M. lutarioriparius).”
L372-373 Briefly state how seed moisture content was measured.
Ans: Thank you very much for your comments. A brief description of how the seed moisture content was measured has been added to the Methods section.
Page 22, lines 385-386
“This was measured by the oven method at 105°C until constant weight was achieved.”
L446 State how many seeds per line were examined.
Ans: We thank the reviewer for raising this point. We have addressed this point in the Methods section. As an exact count of sectioned seeds was not maintained during the extensive histological preparation process, we now state that “Multiple seed samples were prepared and imaged microscopically. The micrographs shown are representative of the observations.”
Page 25, lines 465-467
“Multiple seed samples were prepared and imaged microscopically. The micrographs shown are representative of the observations.”
Reviewer 2 Report
Comments and Suggestions for Authors
The present work, was a work that took long time, 5 years of storage.
The works aimed to analyze 5 cultivars for 0, 4 and 5 years of four different storaged conditions: RT, room temperature, RTD, room temperature with dessicant conditons, LT=low temperature and LTD, low temperature with dessicant conditions.
Why -18 oC?, as a low temperature?, in terms of commercial conditions, maintin -18 oC will be more expensive than 4oC, Why 4oC was not tested?
Table 1, is a big table, than later is simplified in table 2 and fig 2 and even Fig. 3. Maybe Table 1 should be as supplementary Table?
The same for Table 3
Figure 2. shows that germination and vigor index, decreased significant, but then conclusion is better to storage at low temperature? figure 2 is contrary to Fig 3. where Low temperature showed the highest germination
Line 165. seed stored at room temperature no were able to germinate or seedlings died. What this is for?
Figure 5. that means other?
Line 227 (Figure 6A and 6B)
Line 270- 273.
the dessicant´s role, slows down aging and degradation.. but with dessicant failed to germinated?, this is not clear, because aging and degradation is a cause of low germination, but here seems that if seeds are not aged it is not able to germinate?.
line 273. higher relative humicity increases seed moisture content .. then accelerates seed aging and reduces germination...
then it seems that lines 270-274 is contradictory?
Later
Line 277. using desiccants.. allow to maintain the seeds viable even after 5 years.. thus.. is good or not the use of desiccants?
Line 294.. Flores et al. [26]... for a year. Similarly, Meena et al. [27]... quality indicators. Furthermore, Barzali et al. [28] discovered..... year storage experiment. However..
Line 336 (Ratajczak et al...number??)..
Line 340 (Ebone et al. nuber??)
Conclusions should be after the end of discussion, LIne 361
Line390.. 4.58 oC to 30.83 oC.. it was so exact the temperature measured?
Line 423. The paraffin sectioning was performed...
Line 228.. (Figure 6C and 6D)
Author Response
Reviewer #2:
The present work was a work that took long time, 5 years of storage. The works aimed to analyze 5 cultivars for 0, 4 and 5 years of four different storaged conditions: RT, room temperature, RTD, room temperature with dessicant conditons, LT=low temperature and LTD, low temperature with dessicant conditions.
Ans: Thank you very much for giving us this valuable opportunity to revise. Your review comments are very pertinent and instructive, providing us with valuable directions. We have carefully revised the manuscript according to your suggestions. The revised version has been submitted, with all changes highlighted in red for your convenience. Additionally, we have made further improvements beyond the specific comments to enhance the manuscript's quality. We kindly ask for your review of the revised version.
Why -18℃?, as a low temperature?, in terms of commercial conditions, maintin -18℃ will be more expensive than 4℃, Why 4℃ was not tested?
Ans: We thank the reviewer for raising this important point regarding storage temperature and cost. The selection of -18°C was guided by the protocol for long-term seed storage research [1]. This temperature range (-18°C to -20°C) is recommended for orthodox seeds because it virtually halts metabolic activity and maximizes longevity. While 4°C is indeed more economical, it is generally considered suitable for short-term storage (<18 months). Since our study aimed to assess viability over a five-year period, -18°C was the scientifically appropriate choice.
- De Vitis, M., Hay, F.R., Dickie, J.B., Trivedi, C., Choi, J., Fiegener, R., 2020. Seed storage: maintaining seed viability and vigor for restoration use. Restoration Ecology 28, S249-S255.
Table 1, is a big table, than later is simplified in table 2 and fig 2 and even Fig. 3. Maybe Table 1 should be as supplementary Table? The same for Table 3.
Ans: Thank you for your suggestion. As recommended, the original Table 1 and Table 3 have been moved to the supplementary materials and are now presented as Supplementary Table S1 and Supplementary Table S2, respectively.
Figure 2. shows that germination and vigor index, decreased significant, but then conclusion is better to storage at low temperature? figure 2 is contrary to Fig 3. where Low temperature showed the highest germination
Ans: We appreciate the reviewer's careful examination of the figures. The perceived contradiction stems from the different comparisons being made. Figure 2 tracks the change in seed quality over time (a temporal comparison), confirming that seed viability decreases with prolonged storage. Figure 3, in contrast, compares the performance across different storage conditions (a conditional comparison). Our conclusion that low temperature is better is based on the data in Figure 3, which shows that it is better to preserve seed viability relative, despite the overall decline shown in Figure 2.
Line 165. seed stored at room temperature no were able to germinate or seedlings died. What this is for?
Ans: Thank you for your valuable suggestions. The sentence has been revised for clarity and conciseness, as suggested.
Page 10, lines 151-152
“Although some seeds stored at room temperature with desiccant initially emerged, none survived.”
Figure 5. that means other?
Ans: Thank you for your valuable suggestions. The 'Other' category in Figure 5 has been defined in the figure caption as the composite membership value for vigor index, emergence percentage, shoot length, and dry biomass.
Page 13, lines 194-195
“Other, the composite membership value for vigor index, emergence percentage, shoot length, and dry biomass.”
Line 227 (Figure 6A and 6B).
Ans: Thank you very much for your reminder. The suggested revisions have been implemented in the manuscript.
Page 13, lines 206-209
“The cell membranes of embryo and endosperm cells remained intact, with small intercellular gaps and neatly arranged cells (Figure 6A and 6B). In contrast, the seeds stored at room temperature exhibited significant structural damage (Figure 6C and 6D).”
Line 270-273. the dessicant´s role, slows down aging and degradation.. but with dessicant failed to germinated?, this is not clear, because aging and degradation is a cause of low germination, but here seems that if seeds are not aged it is not able to germinate?. line 273. higher relative humicity increases seed moisture content .. then accelerates seed aging and reduces germination... then it seems that lines 270-274 is contradictory? Later. Line 277. using desiccants.. allow to maintain the seeds viable even after 5 years.. thus.. is good or not the use of desiccants?
Ans: We thank the reviewer for this astute observation, which has helped us correct a critical lack of clarity in our original wording. The sentence on lines 270-273 was indeed poorly phrased and created a contradiction. We have rewritten it.
Page 16, lines 250-252
“The successful germination of seeds stored at room temperature with desiccant is likely attributable to the maintenance of low relative humidity, which retards seed aging and degradation.”
Line 294.. Flores et al. [26]... for a year. Similarly, Meena et al. [27]... quality indicators. Furthermore, Barzali et al. [28] discovered..... year storage experiment. However..
Ans: Thank you for your valuable suggestions. The paragraph in question has been thoroughly rewritten to improve the logical flow and coherence of the argument.
Page 17, lines 274-292
“For Miscanthus seeds, vacuum storage demonstrated a limited effect on maintaining seed viability. The vacuum treatment isolating the moisture and oxygen, failed to preserve Miscanthus seeds during the long-term storage, as seeds at room temperature with vacuum lost viability by the fourth year. Moreover, the germination percentage and vitality index of the seeds stored under the low temperature with vacuum conditions were the lowest among the three low-temperature storage methods. This result contrasts sharply with the benefits of vacuum storage reported in other species. For instance, Flores et al. reported that the vacuum treatment prevented a 50% reduction in the germination capacity of Juglans nigra L. seeds stored at -20°C for one year [26]. Similarly, Meena et al. observed that vacuum packaging extended the shelf life of soybean seeds to 18 months without compromising their quality indicators [27]. Furthermore, Barzali et al. discovered a positive effect on the germination percentage of Secale graale L. seeds under the room temperature vacuum conditions during a 26-year storage experiment [28]. Hence, the effectiveness of vacuum storage is highly dependent on species. For seeds such as those of Miscanthus, the vacuum environment may not adequately suppress aging processes, or the seeds themselves might be more sensitive to the vacuum conditions. Therefore, when formulating storage protocols, it is crucial to fully consider species specific seed characteristics, rather than treating vacuum storage as a universally applicable optimization method.”
Line 336 (Ratajczak et al...number??).. Line 340 (Ebone et al. nuber??)
Ans: Thank you very much for your reminder. The suggested revisions have been implemented in the manuscript.
Conclusions should be after the end of discussion, Line 361
Ans: Thank you for your valuable suggestions. The Conclusions section has been moved to its proper position, now placed after the end of the Discussion section.
Line390.. 4.58℃ to 30.83℃.. it was so exact the temperature measured?
Ans: Thank you for your suggestion. The values presented are indeed accurate and represent the mean monthly temperatures calculated from continuous monitoring data, as detailed in Figure S3.
Page 22, lines 400-403
“Detailed monthly fluctuations under the ambient condition are provided in Figure S3. Room temperature storage conditions entailed exposure to the ambient environment of the experimental site. The mean temperature ranged from 4.6°C to 30.8°C.”
Line 423. The paraffin sectioning was performed...
Ans: Thank you for your valuable suggestions. The suggested revisions have been implemented in the manuscript.
Page 24, lines 443-444
“The paraffin sectioning was performed to examine the morphological changes in Miscanthus seeds after storage.”
Line 228.. (Figure 6C and 6D)
Ans: Thank you very much for your reminder. The suggested revisions have been implemented in the manuscript.
Reviewer 3 Report
Comments and Suggestions for Authors
This manuscript presents a comprehensive five-year study examining optimal storage conditions for Miscanthus seeds, addressing an important practical gap in biomass crop propagation. The research is methodologically sound and provides valuable insights into seed longevity factors including temperature, humidity, and genotype effects.
ABSTRACT
- The abstract mentions "practical significance to Miscanthus farmers, seed producers, and storage facilities" but doesn't articulate specific recommendations. Include the optimal storage protocol (−18°C with desiccant) more prominently.
- Specify that evaluations were conducted at 4 and 5 years of storage to give readers a clear timeline of the experimental design.
- Rather than stating findings "could be of practical significance," assert their definitive value and provide a clear recommendation for stakeholders.
INTRODUCTION
- The statement that "Miscanthus seeds do not exhibit dormancy" (line 68) is crucial but appears too late and lacks supporting citations. This fundamental characteristic should be introduced earlier with proper references and explained in terms of its storage implications.
- Only one reference (Meyer et al.) is cited regarding Miscanthus seed storage. Acknowledge this limited knowledge base explicitly and explain why this represents a critical research need given the increasing interest in seed-based Miscanthus propagation.
MATERIALS AND METHODS
- The "room temperature" condition lacks precision. Provide actual temperature ranges recorded during the study (the data in Figure S2 should be referenced here). Specify: (a) mean temperature and standard deviation, (b) maximum and minimum temperatures experienced, and (c) frequency of temperature monitoring. The same applies to the low-temperature condition (−18°C ± what range?).
- Several methodological decisions lack rationale: (a) Why were seeds tested at 4 and 5 years but not at intermediate timepoints (1, 2, 3 years)? (b) Why were only 30 seeds per treatment used? (c) Why was the field trial conducted only in year 5? Provide statistical or practical justifications for these choices, or acknowledge them as study limitations.
RESULTS
- Several important comparisons are not adequately presented: (a) correlation analysis between germination percentage and field survival rate, (b) comparison of vigor index between genotypes within each storage condition at year 5, (c) rate of germination decline from year 0 to 4 to 5 for each condition, and (d) cost-benefit analysis of different storage methods (even if preliminary). Consider adding these as a new subsection "4.6 Comparative analysis of storage effectiveness."
DISCUSSION
- Section 3.5 provides a storage recommendation but lacks: (a) cost comparison of different storage methods (equipment, energy, maintenance), (b) scalability considerations for commercial seed production, (c) discussion of alternative storage technologies (cryopreservation, controlled atmosphere), (d) recommendations for monitoring seed viability during storage, and (e) guidance on how frequently viability testing should be conducted. Add a paragraph addressing these practical implementation issues.
Author Response
Reviewer #3:
This manuscript presents a comprehensive five-year study examining optimal storage conditions for Miscanthus seeds, addressing an important practical gap in biomass crop propagation. The research is methodologically sound and provides valuable insights into seed longevity factors including temperature, humidity, and genotype effects.
Ans: Thank you very much for giving us this valuable opportunity to revise. Your review comments are very pertinent and instructive, providing us with valuable directions. We have carefully revised the manuscript according to your suggestions. The revised version has been submitted, with all changes highlighted in red for your convenience. Additionally, we have made further improvements beyond the specific comments to enhance the manuscript's quality. We kindly ask for your review of the revised version.
ABSTRACT
The abstract mentions "practical significance to Miscanthus farmers, seed producers, and storage facilities" but doesn't articulate specific recommendations. Include the optimal storage protocol (−18°C with desiccant) more prominently. Specify that evaluations were conducted at 4 and 5 years of storage to give readers a clear timeline of the experimental design. Rather than stating findings "could be of practical significance," assert their definitive value and provide a clear recommendation for stakeholders.
Ans: Thank you for your valuable suggestions. In response, we have comprehensively revised the abstract to prominently feature the optimal storage protocol, specify the evaluation timeline, and assert the definitive practical value of our findings with clear recommendations for stakeholders.
Page 1-2, lines 14-30
“Seed storage is critical for preserving genetic resources, but optimal long-term storage conditions for Miscanthus seeds have not been established. In this five-year study, we evaluated storage protocols by comparing seed germination after four and five years, along with field establishment performance. The results demonstrated that genotype, storage conditions, and the storage duration all significantly influenced germination percentage and vigor index of Miscanthus seeds. Low temperature storage yielded the highest germination percentage (59.44%) and vigor index (132.06) in the 4th year, while low temperature with desiccant gave the highest germination percentage (42.41%) in the 5th year. The field performance after direct sowing was also significantly influenced by genotype and storage conditions, with the highest seedling survival rate (7.80%) observed under low temperature with desiccant. The seeds stored under low temperature exhibited minor structural damage, with the intact cell membranes, the small intercellular gaps, and the orderly cell arrangement. Through comprehensive evaluation, storage at -18°C with desiccant was determined to be optimal. Based on these results, we strongly recommend storing Miscanthus seeds at -18°C with desiccant. This protocol offers a reliable and effective solution for farmers, seed producers, and storage facilities to ensure long-term seed viability.”
INTRODUCTION
The statement that "Miscanthus seeds do not exhibit dormancy" (line 68) is crucial but appears too late and lacks supporting citations. This fundamental characteristic should be introduced earlier with proper references and explained in terms of its storage implications.
Ans: Thank you for your kind reminder. The statement regarding the absence of dormancy is an observation from our study. The statement has been moved forward from its original location to an earlier part of the Introduction.
Page 3, lines 53-61
“However, initial germination tests observed that Miscanthus seeds exhibit no dormancy and can germinate immediately after harvest. This characteristic results in spontaneous germination under favorable storage conditions, leading to a loss of seed stocks. Furthermore, germination during storage increases ambient humidity, promoting microbial growth and thereby raising the risk of seed mold and disease incidence. Patanè et al. reported that after one year of storage at room temperature, the germination percentage of Miscanthus seeds significantly decreases from 95.6% to approximately 60% [15]. This decline in germination rate necessitates the use of more seeds for sowing, ultimately increasing cultivation costs.”
Only one reference (Meyer et al.) is cited regarding Miscanthus seed storage. Acknowledge this limited knowledge base explicitly and explain why this represents a critical research need given the increasing interest in seed-based Miscanthus propagation.
Ans: Thank you for your valuable suggestions. We fully agree that the knowledge base on Miscanthus seed storage is currently limited, as correctly pointed out by the scarcity of specific references. We have now explicitly acknowledged this limited knowledge base in the Introduction of our revised manuscript.
Page 3-4, lines 62-72
“The importance of seed storage has been recognized since plant domestication. The deterioration of seeds over time, characterized by declines in morphological structure, physiological function, and biochemical integrity [16], leads to reduced viability and germination failure [17,18]. Inappropriate storage conditions can accelerate this process, with temperature, oxygen, and relative humidity acting as critical factors [19]. For instance, the germination percentage of Ruppia sinensis seeds remained at 74.4% after 9 months of storage under the dry conditions at 5°C, compared to a decline to 35.7% when stored at room temperature for the same duration [20]. Currently, systematic research on storing Miscanthus seeds remains limited. As Miscanthus gains prominence, developing an economical, efficient, and scalable seed storage system has become an urgent requirement for supporting its industrial expansion.”
MATERIALS AND METHODS
The "room temperature" condition lacks precision. Provide actual temperature ranges recorded during the study (the data in Figure S2 should be referenced here). Specify: (a) mean temperature and standard deviation, (b) maximum and minimum temperatures experienced, and (c) frequency of temperature monitoring. The same applies to the low-temperature condition (−18°C ± what range?).
Ans: We thank the reviewer for this critical suggestion to improve the precision of our storage condition descriptions. We have comprehensively revised the Methods section.
Page 22-23, lines 400-405
“Detailed monthly fluctuations under the ambient condition are provided in Figure S3. Room temperature storage conditions entailed exposure to the ambient environment of the experimental site. The mean temperature ranged from 4.6°C to 30.8°C. Maximum temperatures commonly occurred in July and August, with peaks reaching 40.0°C, while minimum temperatures were recorded in January and February, dropping to as low as -5.0°C.”
Several methodological decisions lack rationale: (a) Why were seeds tested at 4 and 5 years but not at intermediate timepoints (1, 2, 3 years)? (b) Why were only 30 seeds per treatment used? (c) Why was the field trial conducted only in year 5? Provide statistical or practical justifications for these choices, or acknowledge them as study limitations.
Ans: Thank you for your suggestion. We have provided the following justifications in the revised manuscript and summarize them here: (a) The germination tests were conducted at the later stages of storage because our primary research focus was on evaluating the long-term effectiveness of the storage methods. A substantial loss of seed viability is theoretically and empirically expected to be more pronounced after several years. By focusing on these critical later time points, we could more effectively discriminate the performance between the different storage conditions, which is the core objective of this study. (b) The use of 30 seeds per treatment with three replications ensures statistical robustness for comparative analysis while respecting practical constraints. This standard practice is supported by our citation of the relevant reference. (c) The field trial was conducted in the fifth year as the definitive test of practical effectiveness. Conducting this trial at the end of the storage period provided a direct and ecologically relevant measure of the ultimate success of our storage protocols for practical application.
Page 23, lines 421-424
“Germination experiments were conducted in 2016, 2020, and 2021, with a focus on the later stages of storage when a substantial loss of viability was expected, to assess the long-term effectiveness of the storage methods.”
Page 23, lines 417-419
“Each treatment per genotype included 30 seeds and was replicated three times to ensure statistical reliability [35].”
Page 24, lines 426-430
“Field trials were conducted in late July 2021 at Hunan Agricultural University to evaluate the practical field establishment potential of seeds after five years of storage. The primary objective was to determine whether the viability observed in laboratory tests after extended storage translated into successful field establishment. This assessment served as the definitive test of the storage method's practical effectiveness.”
RESULTS
Several important comparisons are not adequately presented: (a) correlation analysis between germination percentage and field survival rate, (b) comparison of vigor index between genotypes within each storage condition at year 5, (c) rate of germination decline from year 0 to 4 to 5 for each condition, and (d) cost-benefit analysis of different storage methods (even if preliminary). Consider adding these as a new subsection "4.6 Comparative analysis of storage effectiveness."
Ans: Thank you for your valuable suggestions. We have conducted the additional analyses as recommended, and the results have been incorporated into the revised manuscript as follows: (a) As suggested, we performed a correlation analysis between the final laboratory germination percentage and the field survival rate. This analysis is now presented in Supplementary Figure S1. (b) The vigor index data are provided in Supplementary Table S1. (c) This new quantitative assessment is provided in the Results section. (d) We thank the reviewer for their valuable suggestion regarding a cost-benefit analysis. After careful consideration, we concluded that incorporating a formal economic analysis falls outside the scope and dataset of our current study, which is fundamentally focused on the biological efficacy of storage methods. A rigorous and generally applicable cost-benefit analysis would require extensive data on capital investment, energy consumption, maintenance and seed market prices—parameters that were not collected in our biological experiments. We believe that a preliminary analysis based on our limited data could be speculative and potentially misleading. However, we fully acknowledge the importance of economic considerations for practical implementation and therefore regard this as a valuable direction for future applied research.
Page 10, lines 160-162
“Furthermore, a positive correlation was found between the laboratory germination percentage and the field survival rate after 5 years of storage (Figure S1).”
Page 6-7, lines 117-125
“Seeds stored at room temperature with desiccant exhibited the most severe decline, with losses of 81.61% in germination percentage and 94.07% in vigor index after five years. In contrast, seeds stored at low temperature with desiccant showed the smallest loss in germination percentage (48.63%), while those under low temperature exhibited the smallest reduction in vigor index (60.66%). Furthermore, during the late storage stage (years 4 to 5), the low temperature with vacuum condition was the most effective, minimizing the annual reduction in both germination percentage (12.60%) and vigor index (28.62%).”
DISCUSSION
Section 3.5 provides a storage recommendation but lacks: (a) cost comparison of different storage methods (equipment, energy, maintenance), (b) scalability considerations for commercial seed production, (c) discussion of alternative storage technologies (cryopreservation, controlled atmosphere), (d) recommendations for monitoring seed viability during storage, and (e) guidance on how frequently viability testing should be conducted. Add a paragraph addressing these practical implementation issues.
Ans: We thank the reviewer for these critical suggestions to enhance the practical relevance of our conclusion. We have added a new paragraph in Section 3.5. Regarding point (c) on alternative technologies, we have chosen to focus the implementation discussion on the methods directly tested in our study to maintain clarity and avoid undue speculation.
Page 20, lines 346-354
“To translate these findings into practical implementation, several factors are essential. Although storage at -18°C incurs higher initial and ongoing energy costs than ambient storage, it is strongly recommended for protecting high-value Miscanthus genetic resources, breeding lines, and commercial seed stocks. This investment is offset by a drastic reduction in the need for frequent and costly seed regeneration, making the approach scalable for centralized seed banks and large-scale producers. To ensure sustained viability of Miscanthus seeds, a routine monitoring program is critical. For seeds stored at -18°C with desiccant, we recommend annual germination testing to verify the maintenance of high viability.”
Round 2
Reviewer 3 Report
Comments and Suggestions for Authors
The authors have made substantial and comprehensive revisions in response to Reviewer. The manuscript has been significantly improved both in scientific rigor and practical applicability.
- While the authors appropriately declined to include a speculative analysis, a brief discussion of relative cost factors (energy vs. seed regeneration) could still be valuable
Author Response
The authors have made substantial and comprehensive revisions in response to Reviewer. The manuscript has been significantly improved both in scientific rigor and practical applicability.
Ans: We thank the reviewer for their positive assessment of our revisions and for this constructive follow-up suggestion. We have carefully revised the manuscript according to your suggestions. The revised version has been submitted, with all changes highlighted in red for your convenience. We kindly ask for your review of the revised version.
While the authors appropriately declined to include a speculative analysis, a brief discussion of relative cost factors (energy vs. seed regeneration) could still be valuable.
Ans: Thank you for your valuable suggestions. As recommended, we have added a discussion of the relative cost factors to strengthen the practical perspective of our conclusion.
Page 10-11, lines 321-337
“The results of the comprehensive evaluation indicated that the low temperature with desiccant storage was the optimal for maintaining Miscanthus seeds vitality. This finding is significant for both seed production and long-term preservation, demonstrating that high seed vitality can be sustained using a relatively simple storage method. Although storage at -18°C with desiccant incurs higher energy costs than other storage, it is strongly recommended for protecting high-value Miscanthus genetic resources, breeding lines, and commercial seed stocks. Producing new Miscanthus seeds is a prohibitively expensive and lengthy process, requiring a full year of dedicated land use, controlled pollination, labor-intensive harvesting, and post-harvest processing. By contrast, the energy cost of maintaining a -18°C freezer is relatively modest. Moreover, the desiccant is not a consumable but a reusable resource. It can be regenerated by drying, thereby distributing its cost over many years. Consequently, while storage at -18°C with desiccant has an ongoing cost, it is overwhelmingly cost-effective com-pared to the alternative of frequent seed regeneration. This makes the -18°C with desiccant method not only biologically superior but also economically imperative. To ensure sustained viability of Miscanthus seeds, a routine monitoring program is critical. For seeds stored at -18°C with desiccant, we recommend annual germination testing to verify the maintenance of high viability.”
